# Spatial constraints and stochastic seeding subvert microbial arms race

**Raymond Copeland[1,2]\*, Christopher Zhang [2,3], Brian K. Hammer[3], Peter J. Yunker [1]\***

**1** School of Physics, Georgia Institute of Technology, Atlanta, Georgia, United States of America,
**2** Interdisciplinary Program in Quantitative Biosciences, Georgia Institute of Technology, Atlanta, Georgia, United States of America, **3** School of Biological Sciences, Georgia Institute of Technology, Atlanta, Georgia, United States of America

\* rcopeland38@gatech.edu (RC); peter.yunker@gatech.edu (PJY)

**Data Availability Statement:** The source code and data used to produce the results and analyses presented in this manuscript are available from Github and Handle, respectively: https://github. com/peter-yunker/Spatial-constraints-and-

## Abstract

Surface attached communities of microbes grow in a wide variety of environments. Often, the size of these microbial community is constrained by their physical surroundings. However, little is known about how size constraints of a colony impact the outcome of microbial competitions. Here, we use individual-based models to simulate contact killing between two bacterial strains with different killing rates in a wide range of community sizes. We found that community size has a substantial impact on outcomes; in fact, in some competitions the identity of the most fit strain differs in large and small environments. Specifically, when at a numerical disadvantage, the strain with the slow killing rate is more successful in smaller environments than in large environments. The improved performance in small spaces comes from finite size effects; stochastic fluctuations in the initial relative abundance of each strain in small environments lead to dramatically different outcomes. However, when the slow killing strain has a numerical advantage, it performs better in large spaces than in small spaces, where stochastic fluctuations now aid the fast killing strain in small communities. Finally, we experimentally validate these results by confining contact killing strains of *Vibrio cholerae* in transmission electron microscopy grids. The outcomes of these experiments are consistent with our simulations. When rare, the slow killing strain does better in small environments; when common, the slow killing strain does better in large environments. Together, this work demonstrates that finite size effects can substantially modify antagonistic competitions, suggesting that colony size may, at least in part, subvert the microbial arms race.

## Author summary

Bacterial colonies are often crowded with many bacteria in direct contact. As a result, the competition for space and resources often turns deadly. Bacteria have evolved many mechanisms with which to kill each other; this bacterial warfare is often studied in large communities on agar plates or in flow cells. However, in nature these colonies are often smaller, due to spatial constraints or shear forces. It is unclear how bacterial warfare proceeds in small systems.

stochastic-seeding-subvert-microbial-arms-race
https://hdl.handle.net/1853/73232.

**Funding:** P.J.Y. acknowledges funding from the
NSF Biomaterials (grant no. BMAT2003721), and
NIH National Institute of General Medical Sciences
(grant no. 1R35GM138354-01). B.K.H.
acknowledges funding from the NSF Biomaterials
(grant no. BMAT2003721). The funders had no
role in study design, data collection and analysis,
decision to publish, or preparation of the
manuscript.

**Competing interests:** The authors have declared
that no competing interests exist.

We performed individual based model simulations of bacterial warfare comprising two strains, each capable of killing the other on direct contact. We found that the community size played a substantial role in determining the outcome. When at a numerical disadvantage, the slow killing strain survived at much higher rates in small communities. In fact, there were many conditions in which the slow killing strain survives in small spaces but is completely eliminated in large ones. Conversely, when the slow killing strain is more common, it performs better in large spaces. Together, these observations demonstrate that finite size effects aid the strain that is at a disadvantage, and in some conditions, can even flip which strain increases its abundance.

Finally, we experimentally tested the results of these simulations. Two mutual killing strains of *V. cholerae* were grown unconfined on agar plates (i.e., in large spaces), confined within square holes with sides $7.5 \mu m$ long (i.e., in small spaces), and confined within large square holes with sides $60 \mu$ long to control for effects of the grids. In these experiments we found that the slow killing strain survived at significantly higher rates in small confinement, validating simulation results.

## 1 Introduction

Bacteria in nature are often in close contact and form colonies, crowded, surface attached communities by secretions of an extra cellular matrix [1]. These structures comprise a multitude of different species and strains [2], and thus social interactions are commonplace. These can be direct, such as quorum sensing [3], or indirect, but broadly, these interactions can be categorized based the strains aiding or inhibiting one another: mutualism, parasitism, competition, etc [4]. Competition, where strains inhibit one another, often includes antagonistic interactions [5, 6]. As a result, survival is often a function of killing ability, potentially initiating a microbial arms race [7]. For example, the Type VI secretion system (T6SS), a contact dependent killing mechanism, is found in 25% of gram negative bacteria [8]. However, the effectiveness of the T6SS has been found to depend sensitively on context [9–11], and in many cases the T6SS is not necessary for bacteria to colonize occupied terrain [12, 13]. While it is clear that microbes do not experience a run-away arms race, it is less clear how this race has been subverted.

Spatial structure is known to play an important role in antagonistic competitions [9, 14–18]. Further, environmental structures, such as surface roughness or anisotropic height perturbations, within large communities are known to impact competitive outcomes, and even stabilize coexistence [19, 20]. Crucially, in nature colonies are often spatially constrained by their environment, limiting how large they can grow. These size constraints arise from many different sources, such as environmental barriers and external stresses [21–24]. However, experimental studies of microbial antagonism are typically performed on agar plates or in flow cells, which allow colonies to grow to large sizes, containing thousands of cells or more. It has been previously shown that spatially limited and fragmented cells exhibit different social dynamics and prey cells survive at higher rates in these environments compared to large ones [17, 25] However, little is known about how antagonistic competitions proceed in small environments and with small numbers of cells, despite the natural relevance of such settings.

Here, we use agent-based simulations to explore the effects of spatial constraints on bacterial contact killing, and then validate our observations experimentally. In simulations of mutual killing strains, we find that community size can substantially impact the outcome. When it lacks a numerical advantage over a faster killing strain, the slower killing strain

survives at much higher rates in smaller communities than in larger communities. We show that this effect arises from fluctuations in initial compositions, which are more substantial in small communities. Further, when the slow killing strain has a numerical advantage over a fast killing strain, the slow killing strain is more successful in large communities than in small communities, where finite size effects now benefit the fast killing strain. The outcome of contact killing competitions thus depends on community size. We test these ideas experimentally by inoculating two mutual-killing strains on transmission electron microscopy (TEM) grids, confining bacteria to $\sim 50 \mu m^2$ holes. Antagonistic competitions in TEM grids are highly spatially constrained; we compare these experiments to colonies grown unconfined on agar plates ($\sim 10E6 \mu m^2$) and TEM grids with large holes ($\sim 3500 \mu m^2$). As in our simulations, we experimentally found that the outcome depends both on the community size and initial abundances. When the slow killer is at an initial numerical disadvantage, it performs better in the confined environment; when the slow killer is at an initial numerical advantage it performs better on bare agar. Thus, experiments also demonstrate that community size impacts contact killing competition outcomes.

## 2 Material and methods

### 2.1 Simulation methods

We investigated the impact of community size on contact killing competitions with agent based simulations. We simulate individual cells pushing, growing, reproducing, and killing over time. Our simulations all feature two strains that kill each other on contact; we vary the kill rate of the slow killer to isolate the benefit of contact killing superiority. We perform simulations for many different community sizes, to determine the impact of size on warfare outcomes.

We modeled cells as interacting via a simple harmonic repulsive potential:

$$U = \frac{1}{2}ar^2 \tag{1}$$

where $a = 500$ is the harmonic potential constant and $r$ is the radial overlap between two neighboring cells (Fig 1B). The motion of cells is over-damped, i.e., cells only move when actively experiencing a net force. This approach was inspired by previous work that found that bacterial interactions in bacterial colonies are accurately modeled with stiff, longer ranged interaction potentials with over-damped dynamics [26, 27]. At the beginning of each time step, the order in which cells interact is set at random; as we iterate through the list, we update each cell's position as overlapping neighbors push the focal cell.

Each cell begins with a radius, $R$, of 0.5. Cell mass grows at a constant rate to simulate a homogeneous nutrient distribution, and to isolate the effect of contact killing, rather than growth dynamics and competition for nutrients, on the outcome [6]. We thus model the change in radius per unit time as:

$$\frac{dr_i}{dt} = \frac{G_i}{2\pi R_i} \tag{2}$$

where $G_i$ is the mass growth rate. This approach produces an area that increases linearly with time (i.e., mass increases at a constant rate). Cell division occurs when a cell has doubled in mass, the mother cell spontaneously splits in place into two daughter cells each with half of the maximum mass and with the minimum radius $R_{min}$. To prevent synchronized cellular division across the community, we assign values of $G_i$ to each cell randomly from a uniform distribution with coefficient of variation 0.1 around the mean, $\langle G_i \rangle = \pi R_{min}^2$ [27, 28]. This process takes on average $\Delta t = 1.0$ (one generation) from when an individual cell starts growing to

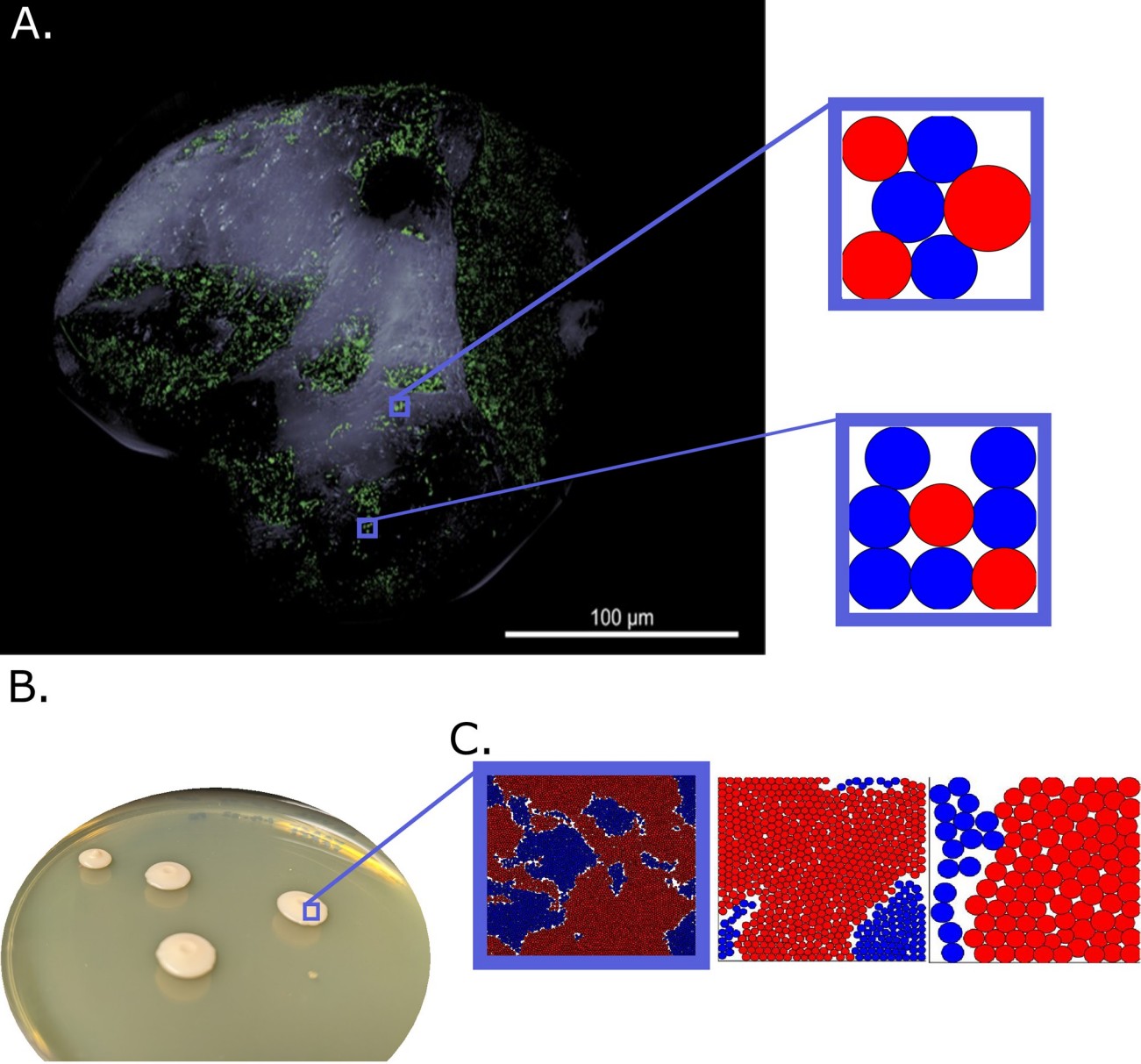

**Fig 1. Microbial communities in nature are often size constrained.** Microbial colonies studied in the lab are often large, such as these *V. cholerae* colonies grown on agar (B.). However, colonies in nature are often much smaller. For example, biofilms on grains of sand (A, adapted from [23]) are significantly smaller than those studied in the lab; we show simulations of $N_{max}$ = 9 cell systems and a $N_{max}$ = 1000 cell simulation with approximate scaling for context. The image in A was used under the Creative Commons Attribution 4.0 International License. C. In our simulations we confine two mutually antagonistic strains (shown in red and blue) to systems of different sizes but with similar cell densities; the shown sizes have ≈ 10, 000 cells, ≈ 1, 000 cells, and ≈ 100 cells respectively after 40 generations of growth. The circles show the extent of the soft intercellular interaction potential.

when it divides. To emulate experimental observations that microbial growth stops under sufficiently large mechanical stresses [29], cell growth in our simulations stops if the system-wide packing fraction exceeds a threshold, $\phi_t$, determined by the system size. $\phi_t$ is determined by:

$$p = \frac{L^2 - (L - 2R_{max})^2}{L^2} \qquad (3)$$

and

$$\phi_t = 0.84(1-p) + \frac{\pi}{4}(p) \tag{4}$$

where L is the length of the system size. These equations weigh the known packing fraction (0.84) that characterizes bulk jammed disks in two-dimensions [30, 31] with the highest packing fraction a disk can reach on a hard edge ($\pi/4$). $p$ is the proportion of space beyond which the largest cells can interact with the edge, which varies with system size. This results in the system either fixating on a single strain at the jamming packing fraction, or, in cases where neither strain fixates, it leads to killing events that consistently reduce the packing fraction below the jamming point for the duration of the simulation, as depicted in Fig J in S1 Text.

Focal cells can potentially kill a cell of the opposing strain if their interaction radii overlap, i.e., if they are in contact. Each strain is assigned a distinct killing rate in units of killing events per time per cell. We will refer to the strain with the lower killing rate as the slow killing strain and the strain with the higher killing rate as the fast killing strain. In all simulations, we hold the killing rate of the fast killing strain, $\kappa_F$, at $\kappa_F = 1$, i.e., on average each of the fast killing strain takes 1 generation to kill an opposing cell (a rate in line with previous works [32]). We vary the killing rate of the slow killing strain ($\kappa_S$, which is always $\leq 1$), enabling us to characterize these competitions by their ratio ($\kappa_r = \kappa_S/\kappa_F$).

The killing probability per time step is the rate multiplied by the length of the time step ($dt = 0.01$), which produces an exponentially distributed "time to kill." If the focal cell contacts its opposing strain, then it may attempt to kill. If the focal cell overlaps with multiple opposing strain cells, one of the opponents is randomly selected for the focal cell to attempt to kill, and the focal cell is allowed to attempt to kill once, regardless of the number of opposing cells that overlap with it. After selecting a target, we generate a random number between 0 and 1; if the random number is smaller than the strain's killing probability, then the focal cell kills the target cell and the target cell disappears.

Cells interact within square environments with varying size. These environments are bounded by stiff walls, modeled with a harmonic repulsive potential with a harmonic potential constant twice that of the cells themselves, thus ensuring cells remain well-confined. We vary the length, $L$, of simulated regions ($L$ = 3, 5, 8, 10, 31, and 100), and we then find the maximum packing fraction of each region according to Eqs 3 and 4 ($\phi$ = 0.817, 0.826, 0.831, 0.833, 0.838, 0.839 for $L$ = 3, 5, 8, 10, 31, and 100, respectively) and then fill the region with the maximum number of cells, $N_{max}$, of minimum radius (0.5) to start a simulation ($N_{max}$ = 9, 26, 68, 106, 1025, 10686 cells for $L$ = 3, 5, 8, 10, 31, and 100, respectively). We place cells into their environment with random positions. To simulate cells randomly attaching to a surface from a planktonic suspension, the proportion of each strain in a given community is determined stochastically. We set the fraction of the slow killing strain in the planktonic suspension, $p_{i,S}$. Upon inserting each cell into the environment, we call a random number between 0 and 1; if it is less than $p_{i,S}$ the cell is set as the slow killing strain, otherwise the cell is set as the fast killing strain. From this stochastic seeding, the initial abundance of slow killers $S_i$ varies from simulation to simulation. We run each simulation for 6400 time steps, i.e., 64 generations, ensuring that all simulations reach a steady state wherein the relative abundance of the two strains changes by less than 0.1%, i.e., no longer significantly changes, over one generation. For each community size ($L$), we simulate many replicates and aggregate the competition results to form a meta-community of many $L$ sized environments.

## 2.2 Experimental methods

We used two strains of *Vibrio cholerae* [33] that use the Type VI Secretion System (T6SS) [34] to deliver toxins to neighboring cells in direct contact. The two strains are isogenic except for their toxins, anti-toxins, and fluorescent proteins. Both strains express Green Fluorescent Protein GFP, but only the slow killing strain expresses Red Fluorescent Protein RFP. The strain growth rates were measured with a BioTek Synergy H1 plate reader and found they were not significantly different: 34.92 ± 0.96 and 35.95 ± 0.87 minutes per generation, where the error is standard deviation and N = 24 each. In all experiments, the two strains were grown separately in LB broth for 24 hours and then mixed together. They were then inoculated with 2*μl* drops on open agar. The slow killing strain was identified by mixing both strains at the same optical density and inoculating on an agar pad; after 24 hours of growth we identified the slow killing strain as the strain that occupied a smaller proportion of space. To confine this competition to small system sizes, we also inoculated bacteria on top of transmission electron microscopy (TEM) grids that were placed on top of LB agar. All steps of all experiments were done at 37˚C. The grids are 3.05 mm in diameter and have over 2000 square holes, each with length and width of 7.5*μm*. Each hole in the grid which can hold ≈ 35 cells in a single layer assuming a high density and using the typical size of a *Vibrio cholerae* [35].

The colonies were imaged on a confocal microscope (Nikon A1R); these measurements returned data for the red, green, and transmitted light channels. Since the two strains differ in fluorescent proteins and toxins, we used confocal microscopy to determine the final proportion of each strain. The proportion of the slow killer was measured in each sample by measuring the number of pixels where the red channel value exceeding the green channel and dividing that number by the total number of pixels that cells occupied; Cells occupied the entire imaged region for the bare agar experiments and only the space in-between the grid lines for the TEM experiments. The transmitted light channel was used to determine which pixels were grid space by binarizing the channel using an Otsu threshold [36]. The pixels that corresponded to the grid were excluded from all calculations.

# 3 Results

## 3.1 Effect of system size on stochastic competitions

To determine the impact of community size on competitions between mutually killing strains of bacteria, we simulated communities in different sized environments ($L$) and strains with different killing ratio $\kappa_r$ and equal initial abundances ($p_{i,S}$ = 0.5). We characterize the outcome by the final relative abundance of the slow killing strain ($\bar{S}_f$), averaged across many simulations, as a function of $\kappa_r$ (Fig 2). We found that in large spaces (i.e., $N$ = 68, 106, 1025, and 10686), the frequency of the slow killing strain is near zero for $\kappa_r$ values less than 0.5. For $\kappa_r > 0.5$, the slow killing population $\bar{S}_f$ quickly increases to 0.5 when $\kappa_r$ = 1.0, as expected for strains with equal killing rates and equal initial relative abundances (Fig 2A). For small environments ($N_{max}$ = 9 and 26), a different trend emerges; the relative slow killer population $\bar{S}_f$ is a linear function of the killing ratio $\kappa_r$ ($R^2$ = 0.991 and 0.946, for $N_{max}$ = 9 and 26, respectively) and $\bar{S}_f > 0$ for most values of $\kappa_r$ we simulated. Examples of the frequency over time dynamics can be found in Fig C in S1 Text.

At this point, it is fair to wonder how much of these results may depend on parameter selection and modeling choices; there are, of course, many other reasonable parameter and modeling choices beyond those described here. We thus explored many more approaches to modeling contact-killing competitions including: periodic boundary conditions, incorporating a relaxation period such that cells start with minimum overlap before cells begin killing and

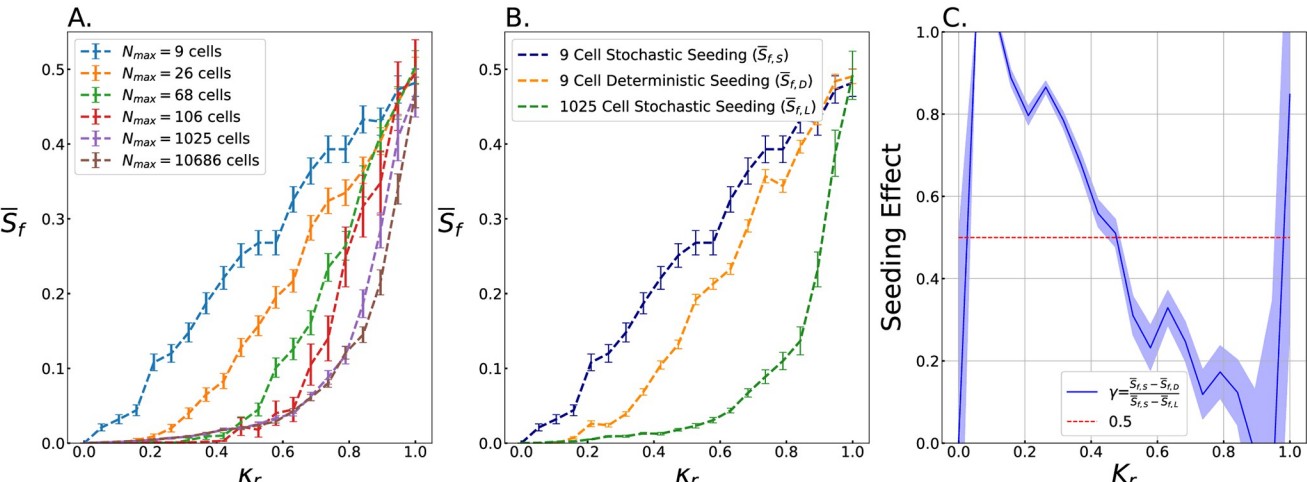

**Fig 2. Slow killing strains survive better in small spaces with equal starting abundance.** A. The fractional mean relative abundance of slow killing cells, $\bar{S}_f$, after 64 generations of simulation is plotted against the relative killing rate of the slow killer, $\kappa_r$, for different size simulations. Each line consists of 20 evenly spaced data points. Each data point is the mean of many simulations; for the $N_{max}$ = 9, 26, 68, 106, 1025, and 10,686 cell simulations we average across 750, 750, 500, 100, 50 and 20 simulations, respectively. For 106, 1025, and 10,686 cell simulations $\bar{S}_f$ decreases rapidly with decreasing $\kappa_r$. Conversely, for $N_{max}$ = 9 cell and 26 cell simulations $\bar{S}_f$ decreases linearly with $\kappa_r$. The standard error in $\bar{S}_f$ from the simulations are shown along each trend. B. $\bar{S}_f$, the final proportion of the slow killing strain, is plotted against $\kappa_r$. Stochastic seeding means that the initial abundance of the slow killing strain was randomly selected from a binomial distribution with equal chance of either strain and the total cell count = 9 ($\bar{S}_{f,s}$), thus modeling random attachment events from planktonic suspension. All simulations in A. are seeded in this manner. Deterministic seeding with an equal proportion is impossible for 9 total cells, so we averaged the trends where the initial abundance of the slow killing strain is deterministically set at 4 and 5 cells out of 9 ($\bar{S}_{f,D}$). There are 20 data points in this trend line; each point is the average of 1000 simulations. These two lines describe simulated "small colony" sizes. The 1025 cell simulations are also stochastically seeded; the initial proportion is expected to be 0.5 +- 0.016. There are 20 data points in this trend line, and each is the average of 50 simulations. This line describes a simulated "large colony" ($\bar{S}_{f,L}$). The error bars represent standard error. C. $\gamma$, the "Seeding Effect" is the difference between the stochastic and deterministic small colony seeding trends divided by the difference between the stochastic, small colony and the large colony trends. This quantifies how much of the "total finite size effect," i.e., the difference between large and small simulations, is a result of stochastic seeding. Here, $\gamma$ is plotted against $\kappa_r$, showing that if $\kappa_r < 0.5$, then $\gamma > 0.5$.

replicating, limiting growth by other means such as a numerical carrying capacity or stopping growth locally for cells that experience a pressure greater than some threshold, as well as starting simulations with all cells at the minimum, maximum, and a random radius. The results we present in this study are robust to these choices; we show the same data for Fig 2A for these other parameter choices in Fig E in S1 Text through Fig I in S1 Text.

To understand the source of the finite size effects observed in Fig 2A, we next investigate the role of stochastically seeding the initial relative abundance of slow killers ($S_i$) in each simulation. To directly test the effect of $S_i$ on outcome ($\bar{S}_f$), we repeated all of the small system size simulations from Fig 1A, but this time set the initial abundance of the slow killing strain deterministically. Deterministic seeding with an equal proportion is impossible for 9 total cells, so we averaged the trends where the initial abundance of the slow killing strain is deterministically set at 4 and 5 cells out of 9. We then compare the results of small systems with stochastic initial conditions, small systems with deterministic initial conditions, and large systems with stochastic initial conditions (Fig 2B). We find that the slow killing strain ends with a higher frequency when the seeding is stochastic for all $\kappa_r$. To quantify this effect, we define the "Seeding effect," $\gamma$, as the ratio of the difference between (stochastic-deterministic) and (stochastic-large cell number), i.e., $\gamma = (\bar{S}_{f,S} - \bar{S}_{f,D})/(\bar{S}_{f,S} - \bar{S}_{f,L})$. We find that the proportion of the effect due to seeding is greatest at low $\kappa_r$. Further, stochastic seeding is responsible for the majority of the difference between large and small spaces when $\kappa_r < 0.5$.

While it is clear seeding plays an important role, the dynamics that lead to this effect are unclear. Therefore, we ran more simulations in which $S_i$ is set deterministically rather than stochastically. We simulated a representative selection of killing ratios $\kappa_r$ and initial abundances $S_i$ for environments with $N_{max}$ = 9 cells and environments with $N_{max}$ = 1025 cells, and we measured $\bar{S}_f$ (Fig 3A and 3C). We found that increasing either $S_i$ or $\kappa_r$ produces an increase in $\bar{S}_f$, as expected (Fig 3A and 3C). Further, individual simulations of small systems typically end with only a single strain surviving, resulting in a large standard deviation in $\bar{S}_f$ (Fig 3B and 3D). In contrast, individual simulations of large systems are more deterministic in their outcomes, thus producing smaller standard deviations in $\bar{S}_f$ (Fig 3D).

To understand the source of fluctuations in stochastically seeded simulations (Fig 2), we next delineated the distribution of initial conditions. We calculated the probability of different $S_i$ values occurring for systems with $N_{max}$ = 9 cells and systems with $N_{max}$ = 1025 cells following a binomial distribution. Every simulation of a given system size starts with the same number of cells; the probability of any cell being a slow killer is 0.5. Thus, the probability of different initial numbers of slow killers is:

$$Prob(S_i) = \binom{N_{max}}{x} p_{i,S}^x (1 - p_{i,S})^{N_{max}-x} \tag{5}$$

where $N_{max}$ = 9 or 1025, $x$ is the initial number of slow killer cells, $S_i = x/N_{max}$, and $p_{i,S}$ is 0.5 as set by well mixed, large planktonic suspension (Fig 4A and 4C). In small spaces, fluctuations can potentially produce a large initial proportion of slow killers (Fig 4A); in fact, $S_i > 0.55$ 50% of the time. Conversely, in large spaces stochastic seeding is unlikely to substantially modify the initial proportion of slow killers compared to the platonic suspension (Fig 4C); in fact, $S_i > 0.55$ only 0.7% of the time.

We next quantified how much each stochastically set value of $S_i$ contributed to survival of slow killers (i.e., $\bar{S}_f$). In other words, we accounted for the fact that larger values of $S_i$ are likely to lead to larger values of $\bar{S}_f$, but are progressively less likely to occur. We thus weighed each outcome of the deterministic simulations (where we set $S_i$) by its probability of occurring from Eq 5 (Fig 4A and 4C) to enable comparisons:

$$w(S_i, \kappa_r) = \bar{S}_f(S_i, \kappa_r) Prob(S_i, \kappa_r) \tag{6}$$

We then normalized these values across rows of constant $\kappa_r$:

$$\tilde{w}(S_i, \kappa_r) = \frac{w(S_i, \kappa_r)}{\sum_{S_i}(w(S_i, \kappa_r))} \tag{7}$$

This calculation represents how much each initial condition contributed to the mean $\bar{S}_f$. We found starkly different results for $N_{max}$ = 9 and 1025 cell simulations (Fig 4B and 4D). For $N_{max}$ = 9, since the total number of cells is relatively small, initial conditions with $S_i > 0.5$ are common (Fig 4A). As a result, a wide range of initial conditions contribute substantially to the final abundance of slow killer cells (Fig 4B). For small environments, we find that relatively rare initial conditions with large $S_i$ can have a substantial contribution. This observation stems from the fact that the slow killer completely is likely to eliminate the fast killer in such conditions, leading to a lottery effect where unlikely initial conditions result in out-sized gains. In particular, for simulations with low relative killing effectiveness $\kappa_r$, slow killers only survive if the initial population $S_i$ is large, so these large $S_i$ initial conditions contribute substantially, despite their rarity. Conversely, in large spaces, the probability of $S_i$ values significantly larger than 0.5 is very small (Fig 4C), so such starting conditions have negligibly small effects.

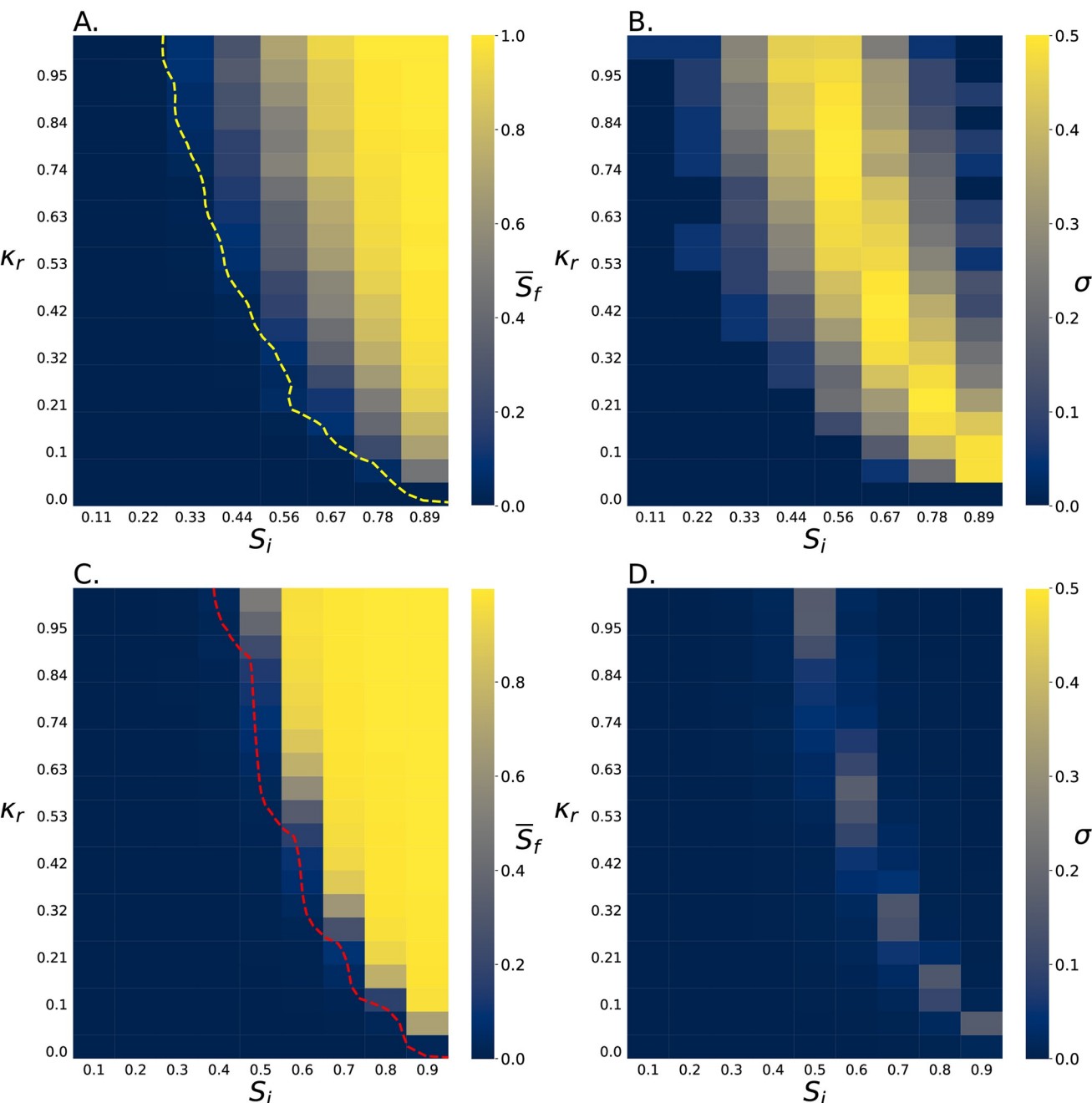

**Fig 3. Slow killing cells in small systems survive in more starting conditions.** A. This heat map shows how the final relative abundance of the slow killing cells, $\bar{S}_f$, depends on its initial relative abundance, $S_i$, and its relative killing rate, $\kappa_r$, in $N_{max} = 9$ cell competitions. Each data point is the result of 1000 simulations. The yellow trend line shows where $\bar{S}_f = 0.05$. B. This heat map shows the standard deviation of $\bar{S}_f$ from simulations in A. The standard deviations are often quite large as competition in $N_{max} = 9$ cell systems typically ends with one strain being eliminated. C. This heat map shows how the final relative abundance of the slow killing cells, $\bar{S}_f$, depends on its initial relative abundance, $S_i$, and its relative killing rate, $\kappa_r$, in $N_{max} = 1025$ cell competitions. In contrast to the $N_{max} = 9$ cell system outcomes, for $N_{max} = 1025$ cell systems, $\bar{S}_f$ is non-zero for a smaller range of $S_i$ and $\kappa_r$. Each of the $N_{max} = 1025$ cell data points is the result of 49 simulations. The red trend line shows where $\bar{S}_f = 0.05$. D. This heat map shows the standard deviation of $\bar{S}_f$ from simulations in C. The standard deviations here are small compared to the $N_{max} = 9$ cell simulations.

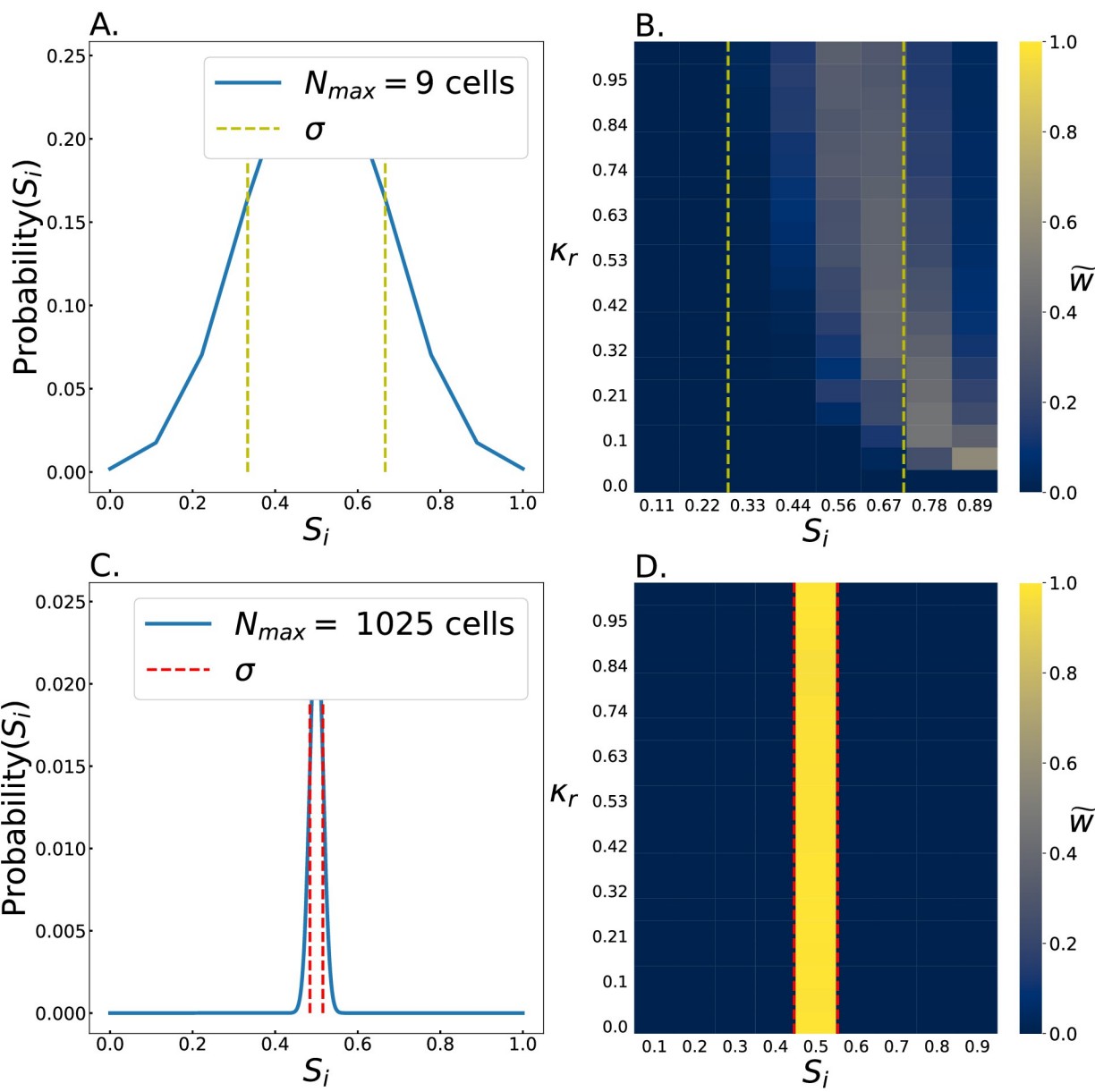

**Fig 4. Sampling fluctuations of initial conditions in small systems favor the slow killing strain.** The probability distribution of the initial relative abundance of slow killing cells, $S_i$, is shown for $N_{max} = 9$ cell and $N_{max} = 1025$ cell systems (A and C, respectively) with on standard deviation outlined for each (yellow for small systems and red for large). The distribution of $S_i$ is of course much broader for the $N_{max} = 9$ cell system than for the 1000 cell system. B, D. These heat maps show the relative impact of each $S_i$ for determining $\bar{S}_f$, as a function of $\kappa_r$. This calculation is done by weighing the values of $\bar{S}_f$ for different $S_i$ from Fig 3A and 3C by the probability that a particular value of $S_i$ occurs (from panels A and C in this figure). This number ($w$) is then normalized across each row, for ease of viewing. In effect, this quantity ($\tilde{w}$) represents how much each $S_i$ contributed to the results seen in Fig 2A. The first standard deviation for panels A and C are shown in B and D for each respective environment.

## 3.2 Effect of system size on invasion

The above analyses show that finite size effects enable slow killing cells to survive at higher rates in small environments than they do in large environments. However, the relative abundance of the slow killer decreases in all conditions studied in Fig 2, as both strains have the

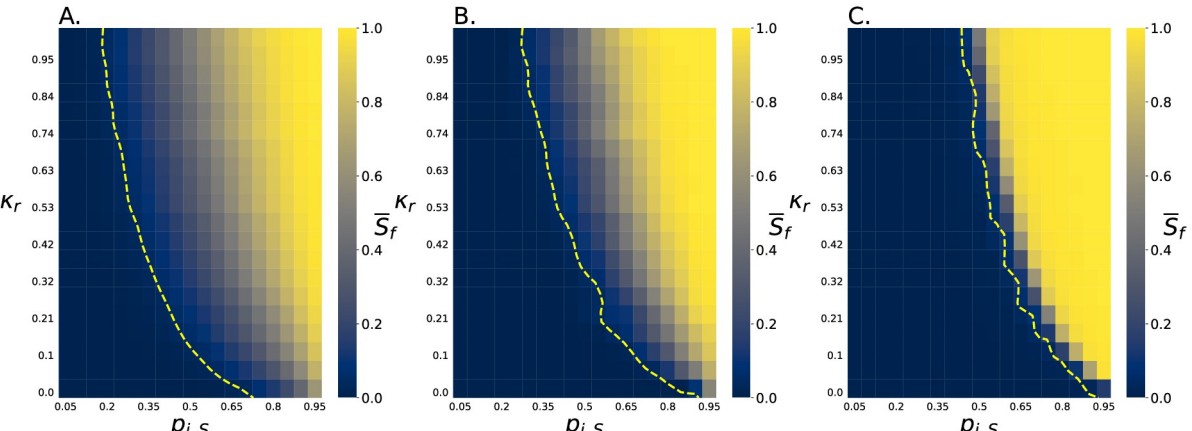

**Fig 5. Slow killing cells survive better in small spaces when stochastically seeded from different planktonic abundances.** A. The final relative abundance of slow killer cells, $\bar{S}_f$, is shown as a function of the relative abundance in the planktonic suspension and the relative killing rate $p_{i,S}$, $\kappa_r$, for $N_{max} = 9$ cell simulations with stochastic seeding. B. The final relative abundance of slow killer cells, $\bar{S}_f$, is shown as a function of the deterministically set initial relative abundance of slow killing cells and $\kappa_r$, for $N_{max} = 9$ cell simulations with stochastic seeding. C. The final relative abundance of slow killer cells, $\bar{S}_f$, is shown for stochastically seeded $N_{max} = 1025$ cell simulations, again as a function of $p_{i,S}$ and $\kappa_r$. In all panels, the yellow trend lines show where $\bar{S}_f = 0.05$. Slow killer cells survive in a much wider range of scenarios in $N_{max} = 9$ cell simulations than in $N_{max} = 1025$ cell simulations, similarly they also survive in more conditions when seeding is stochastic than when seeding is deterministic.

same relative abundance in the planktonic phase and have equal growth rates. We now will focus on the impact of community size when one strain has a numerical advantage over the other strain. These simulations followed a similar protocol as those discussed above, but with different $p_{i,S}$ values (Eq 5, Fig 5A). We found that slow killer cells did in fact survive in more conditions, i.e., for a wider range of both $\kappa_r$ and $p_{i,S}$, in small environments than in large environments (Fig 5A, 5B and 5C). Surprisingly, we found that stochastic fluctuations even help the slow killer survive in small spaces when $p_{i,S} < 0.5$, especially if $\kappa_r > 0.5$. Conversely, slow killers in large spaces are completely eliminated when $p_{i,S} < 0.5$. For example, if $\kappa_r = 0.37$, the slow killer survives in the small environment as long as $p_{i,S} > 0.35$, but only survives in the large environment if $p_{i,S} > 0.6$.

Additionally, this analysis reaffirms that the slow-killing strain exhibits greater survival under stochastic seeding conditions (Fig 5A) compared to deterministic seeding conditions (Fig 5B). Furthermore, it again demonstrates higher survival rates in smaller colonies than in larger ones, as seen in Fig 5C.

We next sought to determine not just the conditions that allow the slow killer to survive, but which conditions actually enable them to increase their relative abundance. We thus plot $\bar{S}_f / p_{i,S}$ (Fig 6A and 6B); when $\bar{S}_f / p_{i,S} > 1$, the slow killing strain increased its relative abundance, and when $\bar{S}_f / p_{i,S} < 1$, it decreased its relative abundance. In contrast to the above results about survival, we found that the slow killer increases its relative abundance in fewer initial conditions in the small environments than the large environments. However, in large environments $\bar{S}_f / p_{i,S}$ decreases sharply once it drops below 1, such that whatever strain is not favored is almost completely eliminated Fig 6. In contrast, small environments exhibit a smaller range of conditions with $\bar{S}_f / p_{i,S} > 1$ and a larger region of space where $\bar{S}_f / p_{i,S} \approx 1$. This means that the outcome depends on a combination of the relative killing rates $\kappa_r$, the invading population size $p_{i,S}$, and community size.

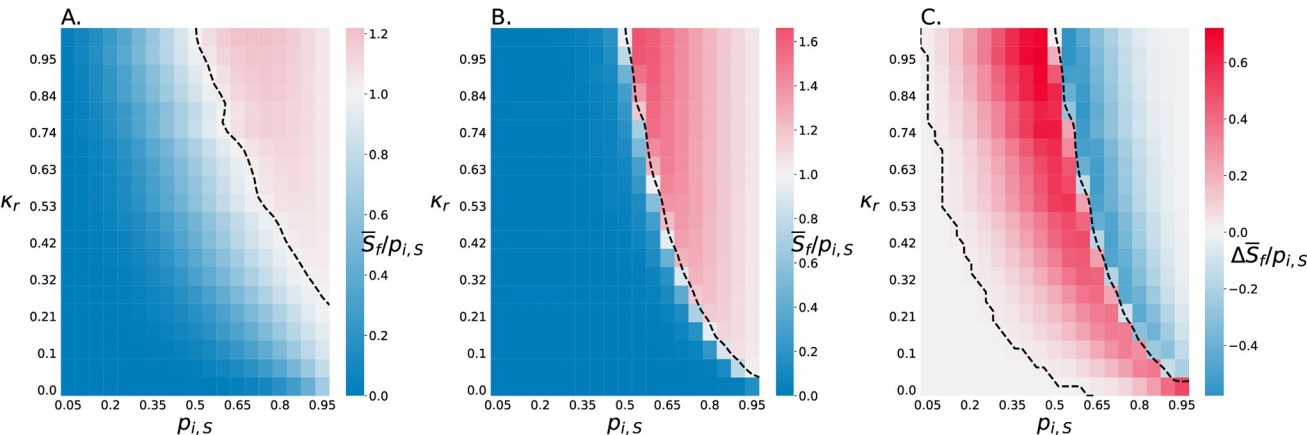

**Fig 6. Slow killing cells are more robust against invasion when drastically disadvantaged.** A, B. These heat maps plot the final relative abundance of slow killing cells divided by the initial relative abundance of slow killing cells in the planktonic suspension, i.e, $\bar{S}_f/p_{i,S}$, as a function of the relative abundance of slow killers in planktonic suspension and the relative killing rate, $\kappa_r$, for $N_{max} = 9$ cell simulations and $N_{max} = 1025$ cell simulations (A and B, respectively). The black trend lines show where $\bar{S}_f/p_{i,S} = 1$. $\bar{S}_f/S_i > 1$ indicates the slow killing strain increases its relative abundance. C. This heat map shows the difference between $\bar{S}_f/S_i$ for the 9 and $N_{max} = 1025$ cell simulations, i.e., it shows the results from A minus the results from B. Red indicates conditions in which the slow killing strain performs better in the $N_{max} = 9$ cell environment, and blue indicates conditions in which the slow killing strain performs better in the $N_{max} = 1025$ cell environment. White space indicates there is no difference between the two environments. The black contour line shows where the difference is equal to 0 indicating there is no difference between the large and small space.

To further explore the impact of community size, we plotted $\Delta\bar{S}_f/p_{i,S}$, the difference in $\bar{S}_f/p_{i,S}$ between small and large environments with identical $p_{i,S}$ (Fig 6C). The same contour that separates where the slow killing strain is less fit than the fast killing strain in the large space (see Fig 6B) separates where the slow killer does better in the small and large environments (Fig 6C). This quantification makes it clear that the slow killing strain survives better in small spaces with conditions that cause it to lose in the large spaces. This is true outside of some edge cases where the slow killing strain starts with too small $p_{i,S}$ or $\kappa_r$ where it does not survive at all. Further, for any condition in which the slow killing strain increases its abundance in the large space, it will increase its own abundance less in the small space.

### 3.3 Experimental validation

The above simulations demonstrate the potential impact of finite size effects on microbial warfare. We now empirically test if these effects can be observed in experiments by growing two strains of mutual killing bacteria in large and small spaces.

Similar to the simulations we describe above, we found that when the competition begins with equal numbers, the slow killing strain ends with larger relative abundance when confined within the small TEM grids than when competing on bare agar (0.366 and 0.214, respectively, $p = 0.02$). In fact, the slow killer strain performs 71.0% better in confinement than out of confinement, comparing the small TEM grids to the bare agar. These observations support the idea that stochastic fluctuations in small environments decrease the benefit of contact killing speed. The proportion of strains was measured 6 hours after inoculation, sufficient time for T6SS killing to have ceased [9].

To control for effects of the TEM grid, we also grew the same strains in TEM grids with spaces 100 times larger than in Fig 7C and 7B. The mean for the larger TEM grid spaces is even lower than bare agar, but the difference is not statistically significant (p = 0.27). Similar to bare agar, the slow killing strain ends with a statistically significantly larger relative abundance

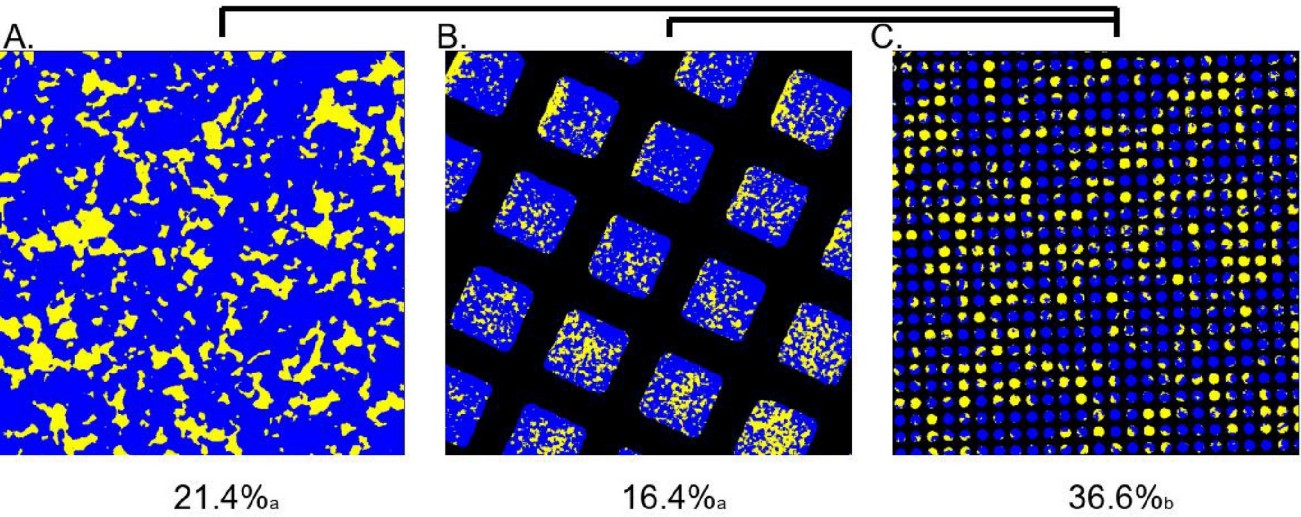

**Fig 7. Experiments validate that slow killing strains perform better in small environments.** A. Two mutual killing strains of *V. cholerae* were grown for 6 hours in large colonies ($\sim 1mm$ in diameter) on bare agar. The strain displayed in yellow is the slow killer and the blue strain is the fast killer. B. The same strains were grown for 6 hours confined to $60\mu m$ square holes in TEM grids. The final fraction of the slow killing strain is not significantly different than that of the bare agar. C. The same strains were also confined to TEM grids with $7.5\mu m$ holes. The final fraction of slow killing cells is significantly larger in these small grids than in either the large grids or bare agar. The mean frequency of the yellow strain is displayed under each panel (N = 5) where subscript show significance groups and connected panels show significant difference ($p < 0.05$).

when confined within the small TEM grids than when confined in the large TEM grids (0.366 and 0.164, respectively, $p = 0.008$). Finally, we note that the slow killer strain performs 123.2% better confined to the small TEM grids compared to the large TEM grids.

Finally, we performed experiments with different initial relative abundances. Specifically, the slow killer began with relative abundances of $p_{i,S} = 0.5, 0.6, 0.7,$ and 0.8. We allowed more time to pass (8 hours) before taking measurements to ensure that all experiments reached a steady state, given that different starting conditions are known to take different amounts of time to reach a steady state [9]. Similar to our simulations, we found that while the slow killer loses significantly when $p_{i,S} = 0.5$ and 0.6, it wins when $p_{i,S} = 0.7$ and 0.8; this observation is true in both large and small spaces (Fig 8A). Further, the strain that loses in the large space always does better in the small space (Fig 8A). When $p_{i,S} = 0.5$ and 0.6, the difference $\bar{S}_f(small) - \bar{S}_f(large)$ is 0.093 ± 0.033 and 0.092 ± 0.031 (mean ± standard error), respectively, indicating that the slow killer has a larger relative abundance in the small environment. When $p_{i,S} = 0.7$ and 0.8, $\bar{S}_f(small) - \bar{S}_f(large)$ is −0.08 ± 0.036 and −0.02 ± 0.009, respectively, indicating that the slow killer has a smaller relative abundance in the small environment (Fig 8B).

## 4 Discussion

Here, we demonstrated that finite size effects can substantially impact the outcome of antagonistic competitions between microbes. In particular, slow contact killing strains survive at a higher rate in smaller spaces than in larger spaces when the two strains initially have equal abundances. We found that in small spaces the final proportion of the slow killing strain is linearly proportional to the ratio of killing effectiveness; this trend persists down to small values of $\kappa_r$. Conversely, in large spaces the final proportion of slow killing cells decreases rapidly as $\kappa_r$ decreases from 1.0, and is essentially zero for $\kappa_r < 0.4$. These phenomena were observed for many different choices for how to construct our simulations, e.g., limiting growth by packing

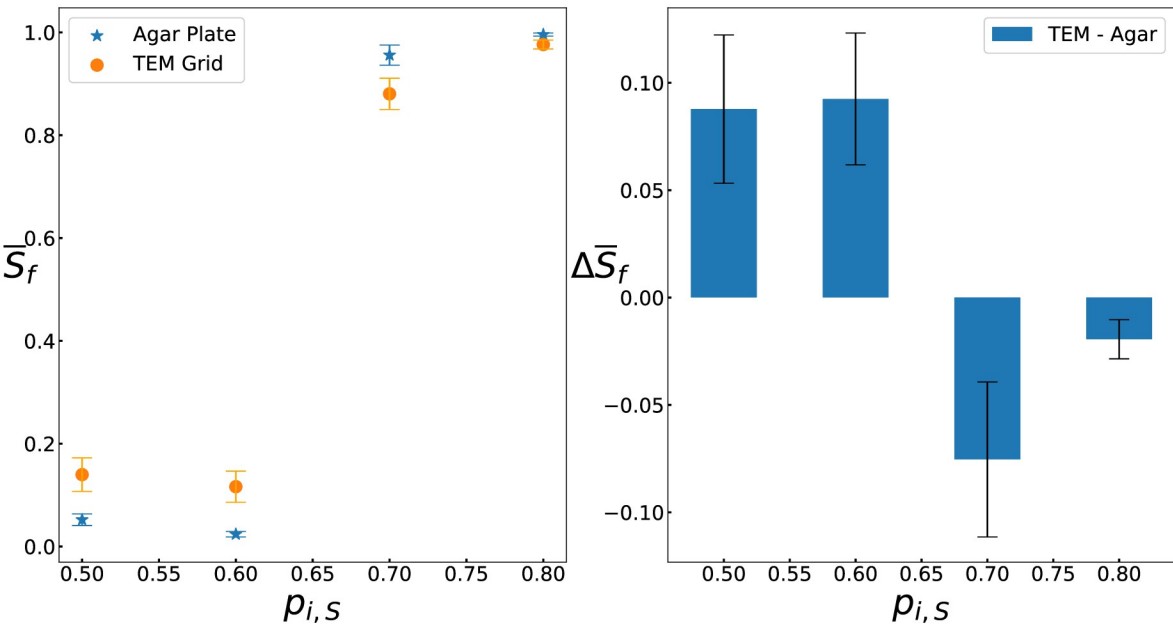

**Fig 8. Experiments validate that small spaces decrease fitness difference across different abundances.** A. Two mutual killing strains of *V. cholerae* were grown for 8 hours in colonies $\sim 1mm$ in diameter on bare agar and in $7.5\mu m$ square holes in TEM grids with different starting abundances ($p_{i,S}$). The means reported are the results of 3,3,6,5 experiments for the open space and 4,3,3,3 for the small grid. Error bars indicate standard error. B. $\Delta \bar{S}_f$, the difference in $\bar{S}_f$ between the small grids and bare agar, is plotted against $P_{i,S}$. These results recapitulate two effects predicted by our simulations. First, for smaller $P_{i,S}$ values where the slow killing strain loses, the slow killing strain performs better in the small system than in the large system (i.e., $\Delta S_f > 0$). Second, for larger $P_{i,S}$ values where the slow killing strain wins, the slow killing strain performs worse in the small system than in the large system (i.e., $\Delta S_f < 0$). Error bars indicate standard error.

fraction, cell number, or local stress. We found that stochastic fluctuations in the initial seeding of these competitions is the primary source of these effects. Further, we found that slow killing cells survive in more conditions when the initial abundance is varied, but the slow killing strain is more likely to increase its abundance in larger spaces. Finally, we performed experiments to validate these simulations by confining bacteria within TEM grids. We observed the same trend in experiments as we did in simulations; the slow killing strain performed much better in confinement than on bare agar when the initial proportions are equal. Thus, these results demonstrate that finite size effects substantially alter antagonistic competitions between bacteria.

The observation that slow killing strains perform better in small environments than in large environments may also have implications for the evolution of microbial warfare. In particular, studies have found both a significant proportion of bacterial strains have contact killing mechanisms [8], and antagonistic interactions are commonplace [6], but there also exists a great diversity of killing ability [11]. These observations seem to indicate that bacteria do not experience an unmitigated arms race in which only the best killing strains can survive. The results presented here suggest that competition in small spaces decreases the value of contact killing speed, which may contribute to the subversion of this arms race. Crucially, when the slow killer has a numerical advantage and "wins" in the large environment, it does not perform as well in the small environment. Finite size effects may mitigate differences between strains in general. Such effects would decrease the fitness advantage of the "superior" strain, constraining its ability to takeover its environment. As a result, finite size effects may lead to more diversity in killing ability.

There are many known mechanisms that limit the utility of microbial antagonism. For example, many studies have found that extracellular matrix production can prevent killing [37]. Rock-paper-scissors game theory can lead to antagonism being a poor strategy. Further, strains can evolve various resistance mechanisms [38–41]. In a different vein, a toxin may not be effective against a novel competitor. However, all of these mechanisms have to do with the behavior of bacteria and their competitors. Here, we highlight that spatial constraints size—a factor that bacteria cannot control—can similarly constrain the benefits of microbial antagonism. Previous work has shown that antagonistic populations mediated by diffusible toxins see a similar effect where spatial structure reduces the invasion potential of stronger antagonist [42], which suggests that the phenomenon investigated here may be general.

Previous publications found that dead cell debris accumulates at the interface between contact killing strains [9, 10], eventually preventing contact killing from occurring. This effect may limit the impact of finite size effects in nature. In large environments, slow killers will do better as eventually dead cell debris will prevent further killing, enabling their survival. In small environments, dead cell debris may not accumulate quickly enough to limit contact killing, but when it does it may limit access to "jackpots," i.e., a single strain completely taking over its environment. However, our experiments and simulations exhibit roughly similar effect sizes (though comparisons are imperfect), suggesting that the finite size effects impact outcomes before dead cell debris prevents contact killing. Further study is necessary to determine which strain, the fast or slow killer, would receive the most benefit in this scenario. Nonetheless, as demonstrated experimentally here, finite size effects do still impact the outcomes of experiments in confinement.

Details about how cells attach to surfaces may play a large role in determining the impact of finite size effects. We model attachment as occurring randomly where each cell has some chance of being either strain and that chance is independent of the cells around it as we assume a well-mixed suspension. However, microbes in nature may form aggregates, which then attach to the surface with a pre-existing spatial structure [43, 44]. These aggregates can be as large as $100um$ which could enhance the impact of finite size effects, as aggregates would increase the size of fluctuations in the initial abundance of a given strain [45]. Conversely, clumping with close relative cells will be less likely to impact the distribution of cells in large systems. In a related vein, while we explored scenarios where many cells attach simultaneously, in nature cells can attach to a surface at different times. This scenario would likely also accentuate finite size effects, as slow killer cells that attach first may have the chance to reproduce before competitors attach. In other words, there may be priority effects that further enhance finite size effects.

Microbes often live in small environments in nature. While the exact proportion of large and small system sizes in nature would be exceedingly difficult to quantify, large open spaces that do not experience significant shear forces are likely exceptional. Thus, understanding how microbial warfare proceeds in nature necessitates understanding how it proceeds in small environments. The results here suggest that differences in fitness between slow and fast killing strains, as well as rare and common strains, are reduced in small environments. Thus, small environments may lead to greater diversity both because of their smaller effective population size as well as because fitness differences are reduced. Future work may try to generalize these results beyond microbial antagonism.

## Supporting information

**S1 Text.** **Fig A**: Finite size effects are independent of $\kappa_F$. **Fig B**: Stochastic Seeding is a major factor in all starting conditions. **Fig C**: 64 generations leads to semi stable frequencies. **Fig D**:

64 generations leads to consistent final frequencies. **Fig E**: Finite size effects are independent of cell growth being limited by low pressure threshold. **Fig F**: Finite size effects are independent of cell growth being limited by medium pressure threshold. **Fig G**: Finite size effects are independent of cell growth being limited by high pressure threshold. **Fig H**: Finite size effects are independent of cell growth being limited by packing fraction. **Fig I**: Finite size effects are independent of cell growth choice being limited by carrying capacity. **Fig J**: Killing events prevent jamming in large systems. **Fig K**: Low Pressure Threshold growth leads to low inter-strain contact.
(PDF)

## Author Contributions

**Conceptualization:** Raymond Copeland, Peter J. Yunker.

**Data curation:** Raymond Copeland, Christopher Zhang.

**Formal analysis:** Raymond Copeland.

**Funding acquisition:** Brian K. Hammer, Peter J. Yunker.

**Investigation:** Raymond Copeland, Christopher Zhang, Peter J. Yunker.

**Methodology:** Raymond Copeland, Brian K. Hammer.

**Project administration:** Brian K. Hammer, Peter J. Yunker.

**Software:** Raymond Copeland.

**Supervision:** Brian K. Hammer, Peter J. Yunker.

**Validation:** Raymond Copeland, Christopher Zhang.

**Visualization:** Raymond Copeland.

**Writing – original draft:** Raymond Copeland, Peter J. Yunker.

**Writing – review & editing:** Raymond Copeland, Christopher Zhang, Brian K. Hammer, Peter J. Yunker.

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
