## [Decision Letter · Decision Letter 0]

14 Aug 2023

Dear Professor Yunker,

Thank you very much for submitting your manuscript "Spatial constraints subvert microbial arms race" for consideration at PLOS Computational Biology.

As with all papers reviewed by the journal, your manuscript was reviewed by members of the editorial board and by several independent reviewers. In light of the reviews (below this email), we would like to invite the resubmission of a significantly-revised version that takes into account the reviewers' comments.

We cannot make any decision about publication until we have seen the revised manuscript and your response to the reviewers' comments. Your revised manuscript is also likely to be sent to reviewers for further evaluation.

Sincerely,

Jacopo Grilli

Academic Editor

PLOS Computational Biology

Rob De Boer

Section Editor

PLOS Computational Biology

Reviewer's Responses to Questions

**Comments to the Authors:**

Reviewer #1: The paper examines the interplay of genetic drift and natural selection in a certain semi-realistic model of spatial growth. The basic result is that in small populations selection is less efficient. Overall, I don't find the results to be novel enough for Comp Bio, but the manuscript could go into PLoS ONE.

Main suggestions:

1. I would supplement each figure with the corresponding plot of the Wright-Fisher model with the matched selection coefficient and population size. This way one could see how much of the effect is "generic" (i.e. the same as in the WFM) and how much is due to the idiosyncrasies of the model including the effect of space and growth dynamics.

2. It would be good to see how frequencies and population size change during the 64 generations at least for some of the conditions.

3. It would be useful to see the distribution of frequencies in the final populations including the number of populations reaching fixation.

Minor:

I couldn't find where the initial population size in each of the simulations is stated.

Reviewer #2: This is an interesting study. In it, the authors study the influence of size constraints and initial conditions on competition between two strains of bacteria that kill each other at different rates upon contact, which arises in many cases in microbiology. To my knowledge a similar study has not been performed before and this paper therefore fulfills the PLoS Computational Biology requirements of originality, innovation, and importance to researchers in the field. It therefore has potential to be publishable. However, there are several concerns relating to the methodology, data analysis, and interpretation of the results that need to be properly addressed before the manuscript can be published.

— A serious concern relates to how mechanical stresses are incorporated in the model. As I understand it (although the description was very unclear), cell growth stops if the global packing fraction exceeds a threshold. But it is well known that the stress can vary greatly spatially in a close packed colony. The stresses go to zero at the edges and compressive stresses build up near the center, which is required from mechanical equilibrium. This spatial variation in stresses which can have important implications for e.g. spatial variations in growth and growth rates is not incorporated at all, which is very concerning.

— Related to the previous question, what exactly is the stress at which growth is stopped at, and how does this compare to experimental results? All the model parameters are given in dimension-free values so comparisons to reality are hard to make.

— A single time step dt=0.01 is used in the simulations. But given the nature in which the contact killing is incorporated in the model, it will be important to show that the results are converged and insensitive to changes in the value of the time step used. This does not appear to have been done.

— Also, it would be useful to show that the results are insensitive to the choice of the shape of the square boundary e.g. use a circle, since a square introduces sharp edges.

— Related to this point: it would be useful to actually show images from the simulations of the spatial distributions of the cells, instead of just abundances. How do these compare to the spatial distribution and sizes of the different aggregates in the experiments? A more rigorous comparison to the experiments would be useful.

— The authors define kr = kf/ks, which should be greater than one, but this appears to have been flipped in the actual presentation of the results.

— In Fig 3, why are the values of the horizontal axis different between A-B and C-D?

— The authors discuss Fig 2 and say "there is a critical value for the killing ratio". Is there? Or is it a gradual increase in the fraction? The authors would do better to be more precise in their language here and in other places.

— The supposed shift shown in Fig 3 is a very weak effect. If anything, it just looks like the boundary becomes more diffuse in the case of the smaller system, as one might expect given the increased importance of fluctuations. Again, it would be useful for the authors to discuss this more carefully and be more precise in their language.

— In the experiments, have the authors verified that the growth rates of the different strains are identical? Otherwise they are changing multiple variables at one time.

— In Fig 7, the authors write 22.2% and 30.3%. What do these numbers mean exactly? What is the uncertainty? Do they really have precision to sub percent resolution?

— In the text discussing Fig 8 the authors write values like -0.048 +/- 0.023. How do they have uncertainty to two significant figures? That does not appear to be meaningful.

— What do they error bars in Fig 8 represent?

— The experiments, while promising, don't show any suitable negative controls. Eg. what happens when you just have one strain growing in each geometry (TEM grid versus without).

— The authors mention biofilms in the abstract and throughout the paper. But it is not clear how the matrix produced by biofilms would influence the results presented here, since presumably that would hinder the contact-mediated killing. This feature is not explicitly considered in the model nor is it clear that the colonies studies in the experiments are actually secreting a matrix (staining would help verify this). It is not clear if the connection to biofilms is even necessary to be made here.

— Minor point: What exactly is shown in Figure 1A? Labels would be useful here. How was this sample prepared? How was it imaged? What color is what? What are the big black circles? Why is the purple diffuse and the green in spots? This image is completely unclear.

Reviewer #3: This is an interesting manuscript that analyzed the effect of population size and random sampling in competitions between antagonist bacteria. The main question, whether small population sizes may alter the outcome of antagonism between such strains, is an important one, given that many microbial populations (e.g., in soil, in the intestine, in skin pores, etc.) are small and spatially separated from others.

I have a few major points that I would like to see addressed before recommending publication:

On the experimental side, I commend the authors for pursuing experimental verification of the results. I am worried, however, that the physiological condition of strains in the two setups may be quite different, possibly leading to differences in the antagonistic interaction dynamics. I would suggest, if possible, to perform also the large-population competitions in TEM grids of different sizes, rather than on agar plates. This would ensure that the bacteria in the two treatments are in comparable states.

On the theoretical side, I wonder how robust the linear trend observed in Fig. 2 for N_max = 9 is to changes in the parameters, most importantly the total simulation time. Intuition would suggest that these curves, especially those for large populations, are not representative of the infinite-time limit, in which due to competitive exclusion we would expect either one or the other strains to completely colonize the landscape, but stochastic sampling of the initial condition may allow \\bar S_f to be different from zero. Is the linear trend observed for N_max = 9 simply due to stopping the simulations early, or is it representative of the infinite-time limit? How do these curves look like in the absence of sampling, i.e. with a fixed S_i = 0.5?

The authors have singled out the effect of sampling by performing simulations with fixed initial conditions, namely the relative abundance of the two strains, but I feel that the cause of the increased survival of the slow killers in small populations hasn't been completely identified. It could be simply due to stochastic fluctuations in small populations, but Fig. 7B makes me wonder what is the effect of space and in particular of the boundary conditions: small populations will have relatively more cells at the boundary, with fewer neighbors than in the bulk, compared to large populations. This might have a big effect in terms of prolonging the survival of the slow killer, especially if found at the boundary. To single out the effect of boundaries, one possibility would be to repeat the simulations with periodic boundary conditions.

It is unclear if the \\bar S_f values plotted in Fig. 3 all come from simulations in which either one of the two populations has fixed (i.e., S_f = 1 or S_f = 0) or not. Does the statement 'there are more initial conditions that enable 175 the survival of the slow killer in the small system than there are in the large system' hold when only data from simulations in which fixation occurred are used? In other words, do stochastic fluctuations (at fixed S_i) allow the slow killer to survive in a broader region of parameter space asymptotically in time, or do they allow it to survive for longer times? Fig. 6 suggests that the latter may be true. These same questions apply to Fig. 5

Overall, the research done is interesting but I think that clarifying the points above should significantly strengthen the manuscript and its conclusions.

Some other minor comments:

On page 8, line 175, the authors state that 'there are more initial conditions that enable 175 the survival of the slow killer in the small system than there are in the large system.' This statement, if I understand correctly, refers to the fact that the region above the dashed white line in Fig. 3A is larger than the region above the red dashed line in Fig. 3C. It seems that the conclusion here would be the same for small values of \\bar S_f, but interestingly it seems as if this is not true for larger values, e.g. \\bar S_f = 0.5. Although I do understand that for this statement a small value of \\bar S_f is appropriate, I thought it would be interesting to compare the relative extent of the regions of parameter space in which \\bar S_f > 0.5 between small and large populations. I would also recommend using the same axes ranges for the four panels, as it would facilitate comparison across panels.

The title of Fig. 4 could be more precise by saying 'Sampling fluctuations of initial conditions in small systems favor the slow killing strain'

At the end of page 8, line 181, does 'these fluctuations' refer to those in Fig. 2, since in Fig. 3 the initial conditions were deterministic? The location of this sentence within the text suggests the opposite, it might be worth specifying which fluctuations are being referred to.

Line 192, 'platonic' should read 'planctonic'

Do the authors see any significant difference between Fig. 5 and Fig. 3? I thought a comparison would be warranted.

The back-and-forth between sampling and non-sampling the S_i distribution is at times a bit confusing and reader is left wondering which of the two are being discussed. It would help more clearly identify which figure

refers to which situation, or discussing deterministic S_i first in the text, and then the sampling

Lines 230-231: here it would be helpful to have a comparison with deterministic, fixed initial conditions.

**Have the authors made all data and (if applicable) computational code underlying the findings in their manuscript fully available?**

Reviewer #1: **No: **I didn't see references to code or data in the manuscript

Reviewer #2: **No: **The github URL provided requires a user id and password to access, neither of which is provided.

Reviewer #3: **No: **I was unable to open the GitHub link provided. It might be technology ignorance on my side.

PLOS authors have the option to publish the peer review history of their article (what does this mean?). If published, this will include your full peer review and any attached files.

Reviewer #1: No

Reviewer #2: No

Reviewer #3: No
---

## [Decision Letter · Decision Letter 1]

8 Jan 2024

Dear Professor Yunker,

We are pleased to inform you that your manuscript 'Spatial constraints and stochastic seeding subvert microbial arms race' has been provisionally accepted for publication in PLOS Computational Biology.

Best regards,

Jacopo Grilli

Academic Editor

PLOS Computational Biology

Rob De Boer

Section Editor

PLOS Computational Biology

Congratulations on the acceptance! One of the reviewers has pointed out that the experimental data to reproduce the results of the paper are not available in the repository https://github.com/peter-yunker/Spatial-constraints-and-stochastic-seeding-subvert-microbial-arms-race . Please update it as required by the journal policy https://journals.plos.org/ploscompbiol/s/data-availability

Reviewer's Responses to Questions

**Comments to the Authors:**

Reviewer #2: The authors have adequately addressed the reviewer comments, and the manuscript is now suitable for publication. Congratulations to the authors.

Reviewer #3: I recommend the revised manuscript for publication: The authors have responded to all my concerns and I think that the new experiments significantly strengthen the conclusions. As a minor note, the authors may want to reference the fact that previous work with well-mixed antagonistic populations has also seen the role played by stochastic seeding in the outcome of antagonistic competition (see Fig. 3D of doi.org/10.7554/eLife.62932), although in different settings than here (not spatially explicit and with antagonism mediated by diffusible toxins), which suggests that the phenomenon investigated here may be quite general and important.

**Have the authors made all data and (if applicable) computational code underlying the findings in their manuscript fully available?**

Reviewer #2: Yes

Reviewer #3: **No: **Upon accessing https://github.com/peter-yunker/Spatial-constraints-and-stochastic-seeding-subvert-microbial-arms-race, I only see the code used to run the simulations, not the experimental data

PLOS authors have the option to publish the peer review history of their article (what does this mean?). If published, this will include your full peer review and any attached files.

Reviewer #2: **Yes: **Sujit Datta

Reviewer #3: No

---

## [Editor Report · Acceptance letter]

23 Jan 2024

PCOMPBIOL-D-23-00945R1 

Spatial constraints and stochastic seeding subvert microbial arms race

Dear Dr Yunker,

I am pleased to inform you that your manuscript has been formally accepted for publication in PLOS Computational Biology. Your manuscript is now with our production department and you will be notified of the publication date in due course.

With kind regards,

Anita Estes
